# Coming Out Strategies on Social Media among Young Gay Men in Malaysia

**Collin Jerome \* and Ahmad Junaidi bin Ahmad Hadzmy**

Faculty of Language and Communication, Universiti Malaysia Sarawak, Kota Samarahan 94300, Malaysia; ahajunaidi@unimas.my
\* Correspondence: jcollin@unimas.my

**Abstract:** Social media have afforded many young gay men the opportunity to explore their identities and practice coming out. The ease of use and the anonymity that can be assumed online has enabled young gay men to come up with different strategies of self-disclosure in social media. The present study seeks to examine these strategies among young gay men in Malaysia, given the limited data on the social media experiences of gay men in the country. In-depth interviews were conducted with six young, gay-identified men to find out the strategies they employed in disclosing their sexual identity online. The analysis of the interview data revealed that the coming-out strategies among the participants mostly resembled those employed by gay men in Western societies, including being out and proud, being out and discreet, and being closeted on social media. The analysis also revealed that the participants held different views regarding the role of social media in the coming out process for gay men in Malaysia. These findings have implications that are relevant to the issues of identity formation among gay men in contexts where homosexuality is still subject to social, legal, and religious condemnation.

**Keywords:** gay; coming out; strategies; social media; Malaysia



## 1. Introduction

Coming out is an important part of the lives of many gay, lesbian, bisexual, and transgender, and queer or questioning (LGBTQ) individuals. It is often referred to as a process involving a single event or a series of events through which individuals (1) accept their sexual orientations and/or gender identity as LGBTQ, and/or (2) tell others about their sexual orientation or gender identity. These events are often described through "the closet" metaphor, wherein LGBTQ individuals liberate themselves from the closet by being true to their sexual orientation and/or gender identity, being honest with others, making connections, and gaining support within their community [1].

Coming out is also often viewed as a lifelong process, with said events taking place at any stage of an individual's life. Such a view has long been the subject of much discussion and debate in the scholarship on coming out, particularly regarding how the process is understood in terms of the developmental stages an individual goes through to achieve an integrated LGBTQ identity [1]. The present study aims to add to this discussion by examining coming out strategies on social media among young gay men in Malaysia.

### 1.1. Coming Out among Young Gay Men

Coming out among young LGBTQ individuals, particularly gay men, has been a subject of investigation for many years, with researchers examining various aspects of coming out, including factors and reasons for coming out, the stages of coming out, the consequences of coming out, and coming out processes and experiences, to name just a few [2–4]. Some of the factors and reasons include coming out to oneself (i.e., internal/personal factor) for self-acceptance, and coming out to other individuals (i.e., external

factor) for their acceptance, validation, and affirmation [2–4]. The consequences of coming out may vary, ranging from positive, liberating experiences to misunderstanding, discrimination, bullying, and harassment [2–4]. The coming out process may be more difficult for different people with different intersections, as they may face greater prejudice and stigma associated with being different from the norm. LGBT individuals in Malaysia, for instance, especially those from the Malay Muslim community, are often ridiculed for being *songsang* (abnormal/inverted) and are rejected for being un-Malay and un-Islamic because non-normative forms of sexuality and gender contravene the religious and cultural values of the Malays [5,6].

Recent years, however, have seen a proliferation of research on coming out among young gay men, including the examination of their coming out experiences on social media. According to Kaplan and Haenlein (2010), "social media is a group of Internet-based applications that build on the ideological and technological foundations of Web 2.0, and that allow the creation and exchange of User Generated Content" [7] (p. 61). Examples of social media include wikis, RSS feeds, blogs, and electronic social networks (e.g., Facebook and Twitter) [8], as well as mobile dating and hook-up apps (e.g., Tinder and Grindr) [9]. The creation and exchange of user-generated content are made possible by social media affordances (e.g., actions or features made available through social media browsers or mobile application apps), which include (1) communicability/interactivity (i.e., social media enable the users to communicate/interact with others), (2) visibility (i.e., social media enable users to view public discussions, contributions, network messages and connections, and profile information), (3) anonymity (i.e., social media enable users to express their opinion anonymously), (4) editability (i.e., social media enable users to craft and recraft their communicative acts), (5) association (i.e., social media allow users to make the connection between individuals, organizations, and actions), and (6) generativity (i.e., social media allow users to express their ideas and experiences) [10].

The study by Owens (2017), for instance, revealed that young gay men (aged between 18 and 27) used various strategies to manage the disclosure of their sexual identity on social media platforms (i.e., Facebook). Such strategies include (1) "Out and Proud" (e.g., gay men used Facebook to acknowledge and assert their sexuality while diligently coming out to others online); (2) "Out and Discreet" (e.g., gay men used Facebook to evasively come out to some people while concealing their sexual identity from others); and (3) "Facebook Closeted" (e.g., gay men used Facebook to laboriously manage their profiles to prevent their sexuality identity from being exposed) [11] (p. 431). It was also observed that Facebook was both emancipatory and precarious for gay men, for two reasons: first, it served as an avenue through which coming out could be enacted, and second, it encouraged gay men to use different strategies to disclose and manage their sexual identity online [11].

In a study by DeVito, Walker, and Birnholtz (2018), they posited that young gay men—along with lesbians, bisexuals, transgender people, queer, and other gender and sexual minority identified individuals (LGBTQ+)—not only used social media for self-presentation but also that they did so with certain goals in mind, and employed specific strategies to achieve them [12]. These goals include "the relatively unfiltered presentation of one's day-to-day life" and "not [wanting] to 'rock the boat' among their followers", while the strategies to achieve these aims include "tagging key individuals" to increase visibility and "heavy privacy tool use" [12] (p. 5). Some LGBTQ+ individuals would tactfully disclose information about themselves on social media, allowing them to stave off stigmatization while retaining avenues for the exploration and expression of their gender and sexual identities [12].

Similar findings were reported by other studies, including that of Hanckel, Vivienne, Byron, Robards, and Churchill (2019). They contended that gay men, along with lesbians, bisexuals, transgender people, intersex people, and other non-heterosexual and gender diverse (LGBTIQ+) young people employed the identity curation strategy in disclosing and managing their self-presentation on social media (i.e., Facebook) [13]. This strategy was enacted through the young people's digital literacy and the functions afforded by social

media platforms that enabled them to disclose certain information about themselves online to a carefully selected list of friends, while blocking, unfriending, and unfollowing others from accessing this information [13]. This identity curation strategy brings benefits in that it opens up many possibilities for LGBTIQ+ young people, including (1) "[allowing] them to speak to specific audiences about topics related to gender and sexuality, particularly on platforms such as Facebook, which provide functionality that allows for engaging with multiple audiences" and (2) "[allowing] for building and fostering supportive peer networks to engage in selective disclosures of gender and sexuality, as well as building support for 'coming out' among peers" [13] (pp. 10, 15).

One other study that is worth mentioning here comes from Duguay (2016), who investigated how LGBTQ individuals negotiate sexual identity disclosure on mainstream social media platforms such Facebook based on their personal experiences with context collapse—a situation or an event through which they intentionally redefined their sexual identity across audiences and managed unintentional disclosure (i.e., sexual identity expressions which became visible to unintended audiences) [14]. The findings reveal that participants took advantage of the different affordances of Facebook to create two key preventative strategies or measures to avoid context collapse: tailoring identity performances and separating audiences [14]. The first strategy allowed the participants to adjust, modify, or customize their sexual identity expressions in ways that would be acceptable to different kinds of audiences. The participants made use of the affordances of Facebook in "maintaining the ambiguity of potential sexual identity indicators through humor, such as changing one's relationship status to being 'married' to a best friend, or by posting messages that heterosexual allies or groups with certain political beliefs also shared" [14] (p. 23). The second strategy enabled the participants to segregate the different audiences for identity performances by using Facebook's affordances to "establish boundaries of varying fortitude depending on the sensitivity of the information being shared and the particular audience", in addition to "tailoring their privacy settings and 'friending' practices within Facebook" [14] (p. 26).

While the studies reviewed here are largely informed by research on LGBTQ individuals of Western backgrounds and in Western contexts, more needs to be known about LGBTQ people from predominantly Muslim countries where homosexuality remains unacceptable. Global surveys conducted by the Pew Research Center in 2014 and 2019 revealed that the acceptance of homosexuality remained low among adherents of Islam in countries with large Muslim populations [15,16]. This may explain the coming out experiences among gay men living in these countries, as evidenced by their various acts of self-disclosure both online and offline. In a study by Triastuti (2021), Indonesian gay men employed several strategies to assert and express their homosexual identity online to overcome stigma and social exclusion in the country. This is mainly because HIV/AIDS is viewed by the larger society as a disease of gay men, and because of how the local dominant structures (e.g., regulatory bodies, schools, media) regard Indonesian gay men as social evils who spread HIV/AIDS [17]. The strategies, among others, include queer literacy (e.g., how Indonesian gay men learn to talk about and understand sexuality and self-disclosure online) and social community activism (e.g., how Indonesian gay men build a powerful peer network to promote sexual health among the gay community, and to produce knowledge and awareness about HIV/AIDS) [17].

Another study by Syahputra and Yuliana (2016) found that gay men in Indonesia used social media platforms (i.e., Jack'd) as an alternative means of communication with other gay men who face difficulties interacting openly with each other. This is because gay men in this Muslim-majority country are discriminated against because of their sexual orientation, which often leads to restricted communication among them [18]. Syahputra and Yuliana (2016) further contended that gay men in Indonesia used Jack'd for social and sexual purposes, which were achieved through several strategies related to the disclosure of their sexual identities [18]. These include being open about their sexual orientation and giving more specific descriptions of themselves online that were needed to develop connection

and chemistry among them. However, despite the affordances of Jack'd (and other social media platforms) for gay men in Indonesia, their online disclosure was not removed from their feelings of enthusiasm and skepticism over online same-sex communication, and offline, face-to-face, interpersonal communication often remained the end goal for these gay men to further develop their same-sex relationships [18].

### 1.2. The Current Study

The findings of the above-mentioned studies on gay men in Indonesia have implications for researchers who wish to further extend the line of inquiry to include gay men from another Muslim-majority country, Malaysia. This is because being LGBTQ in the country is not an easy task, as non-normative sexualities and genders are subject to social-legal condemnation [19–21]. LGBTQ individuals from the Malaysian Malay Muslim community, for instance, regularly experience difficulties in creating and expressing their sexual and gender identities, largely because the Malaysian representation of Sharia laws forbid same-sex sexuality, and because LGBTQ individuals are generally perceived as un-Malay and un-Islamic [5,6] Over the past several decades, Malaysian authorities have become more public and sterner in their stance towards LGBTQ individuals and communities by enforcing various laws and specific legislations (e.g., civil and Sharia) [22,23]. Such laws not only have dire consequences for the lived experiences and material conditions of LGBTQ individuals in Malaysia but also their sense of self and belonging in a country they call home [24–26]. An examination of their coming out experiences on social media may provide insights into the various aspects of these experiences, including the types of social media platforms used for coming out and the strategies employed to disclose and manage sexual identity, given in the context in which they live their lives as LGBTQ. The study by Mohammad Tuah and Mazlan (2020) is worthy of mention here [26]. They explored how Twitter functioned as a safe space for self-disclosure among Malaysian LGBTQ youths, with data gathered from interviews with 10 LGBTQ-identified individuals. The results show that Twitter served as a safe avenue for participants to disclose their sexual identity, with four factors that motivated them to do so:

(1) Knowledge (e.g., the participants were able to share information about being LGBTQ by disclosing their sexuality on Twitter and enhancing their understanding of the community).

(2) Private life (e.g., the participants were able to discuss aspects of their personal life, such as romances and lifestyles, more openly and freely on Twitter).

(3) Self-acceptance (e.g., the participants were able to be themselves on Twitter by coming to terms with their sexuality and coming out to accept their authentic selves without fear of the judgment of others).

(4) Movement and clique (e.g., the participants were able to form LGBTQ-friendly cliques, i.e., virtual groups that help to spread awareness about the LGBTQ community and challenge LGBTQ-related discrimination. Members of the group would respond to anti-LGBTQ tweets and share information about prejudices against the LGBTQ community as a positive show of support for LGBTQ people).

This study is important because of the limited data on social media experiences among LGBTQ individuals in Malaysia. Our study builds upon and extends said research by examining social media use among Malaysian gay men regarding their coming-out strategies on both mainstream social media platforms and gay dating apps.

### 1.3. Theoretical Framework

It is important to note that the study presented in this article formed part of a larger piece of research that examined identity construction on social media among LGBTQ individuals in present-day Malaysia. This research was theoretically informed and guided by Erving Goffman's theory of self-representation, which has thus far foregrounded much research on online identity management on social media. According to Merunková and Šlerka (2019), the theory posits that when we interact with others, they often tend to

control the image that they create inwardly about us and the impressions we make on them. This tendency is realized through their deliberate adjustment or modification of the façade (social front) that functions as the stage for the interaction and the personal façade (personal front) that comprises our outlook and mannerisms [27]. Although the theory applies to the context of offline, in-person interactions, it can also be applied to online, in-person interactions: the façade of such interactions (e.g., social networks) is enacted through the user's interface (i.e., online profiles) and their façade, which is made up of their profile pictures and how they communicate with others on or through their social networks' platforms [27]. The same can be said of LGBTQ social network users who create their online identities through the assemblage of the façade and their façade. More importantly, they disclose and manage their online identities for different goals and through different strategies and management tools, with different benefits and drawbacks depending on the contexts in which they live or find themselves. These include, among others, (1) the exploration and development of their identities that are facilitated by social media, (2) the concealment of authentic selves on social media for impression management, (3) the protection of their identities on social media to manage multiple selves, and (4) the expression of authentic selves on social media [28]. Figure 1, below, illustrates how the above-mentioned theory was developed within the larger research to examine the subject under study.

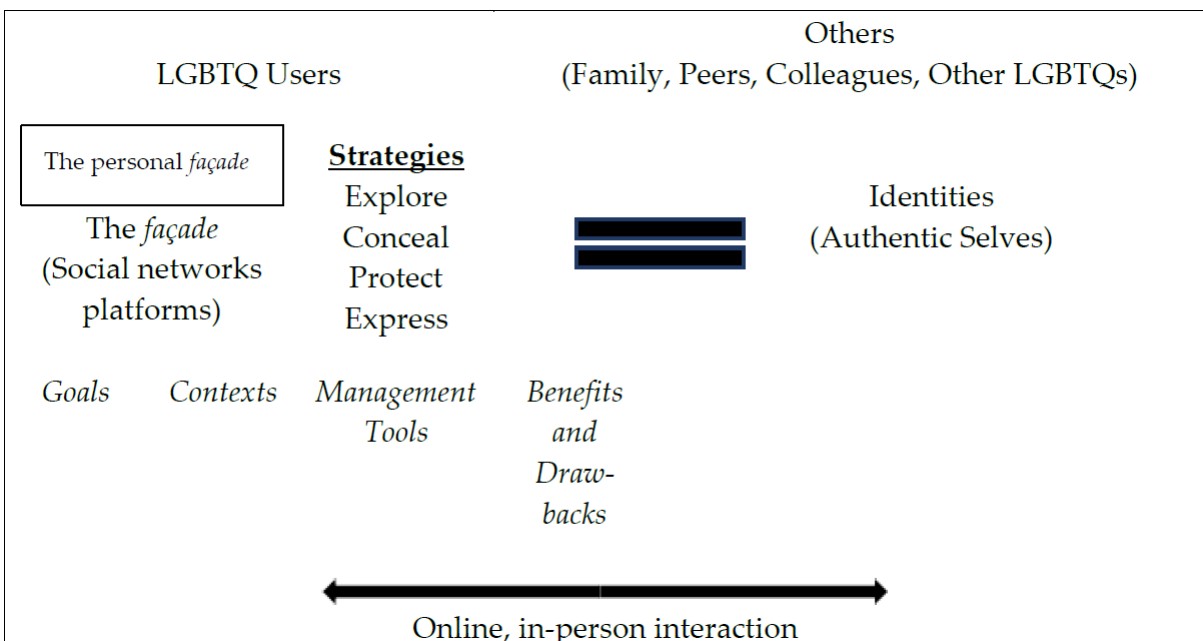

**Figure 1.** Theoretical framework on online identity management on social media.

## 2. Materials and Methods

### 2.1. Participants

The participants were 6 gay-identified men who were based in major cities in Malaysia, namely Kuala Lumpur and Kuching, at the time of the study. These cities were chosen because of the high population of gay-identified men living there. Although the exact amount is difficult to establish, previous studies reported that gay-identified men can mostly be found in the country's capital cities, such as Kuala Lumpur, Penang, Johor, and Kuching, with a majority being from the middle or upper-middle classes [29,30]. The respondents were recruited via chain-referral methods, which are often used to contact hidden populations such as sex workers and people living with HIV/AIDS. These methods refer to "a set of sampling techniques that relies on people who known about a population or are members of that population to gain access to information about the group" [31]

(p. 281). The sample size follows Daniel's (2012) recommendation on the "typical sample sizes for various types of research design"—in this case, between 6 and 10 participants for phenomenological research that uses non-probability sampling [32] (p. 243). The participants were aged between 20 and 30, well-educated, and represented the different religious faiths and ethnic groups in Malaysia. We used the following pseudonyms to ensure the confidentiality and anonymity of the participants, and the ethical obligation that has formed a core part of many research studies on LGBTQ individuals and communities [33–35]. Table 1 provides further information on the participants' demographic information.

**Table 1.** The demographic background of the interview participants.

| Participants | Age | Ethnicity | Religion |
|---|---|---|---|
| Arif * | 23 | Malay | Muslim |
| Azrul * | 27 | Malay | Muslim |
| Dennis * | 27 | Chinese | Buddhist |
| Nas * | 21 | Malay | Muslim |
| Wang * | 30 | Chinese | Buddhist |
| Jad * | 30 | Iban | Christian |

* Pseudonyms.

## 2.2. Ethical Considerations

This study was part of a larger research project funded by the Ministry of Education of Malaysia under the Fundamental Research Grant Scheme (grant number: F09/FRGS/1874/2019). Ethics approval was granted by the Human Research Ethics Committee (HREC) at Universiti Malaysia Sarawak (protocol code: HREC(BP)2020(1)/02; approval date: 8 December 2020). An important ethical consideration made for the study was to support the anonymity of the participants and the confidentiality of the people in their contexts— no identifying information was requested, and any identifying information offered was removed. Pseudonyms were used in all of the direct reporting of quotes.

## 2.3. Data Collection and Analysis Procedures

The Interpretative Phenomenological Approach (IPA) was employed to understand the coming out experiences on social media among gay men in present-day Malaysia. We felt that the approach was appropriate given its aim "to explore in detail how participants are making sense of their personal and social world" [36] (p. 53). More importantly, IPA is phenomenological both in scope and nature for three reasons: first, "it involves a detailed examination of the participant's world"; second "it attempts to explore personal experience, "and third "[it] is concerned with an individual's perception or account of an object or event, as opposed to an attempt to produce an objective statement of the object or event itself" [36] (p. 53). Furthermore, we employed semi-structured interviews as the primary data collection instrument for this IPA-based study. This is because such a method "allows the researcher and participant to engage in a dialogue whereby initial questions are modified in the light of the participants' responses and the investigator can probe interesting and important areas which arise" [36] (p. 57).

Institutional Review Board approval was obtained, as was informed consent before the data collection. The interviews took place online and were conducted in both English and Malay. At the start of the interview, we explained the full purpose of the study and assured the participants of confidentiality regarding disclosed information. We then explained that their involvement was voluntary and that they could withdraw from participation at any time or point during the interview. Recording began once consent was obtained and continued throughout our conversations with the participants. The interview followed the IPA semi-structured interviewing process: the researchers (1) established rapport with the respondents, (2) paid less attention to the ordering of the questions, (3) probed interesting areas that arose during the interview, and (4) gauged the participants' interests and concerns [36] (p. 58). The interview questions were constructed based on the IPA

questioning techniques, including the use of "prompts" and "funneling", as well as neutral, open-ended, and less-technical questions [36] (pp. 62–64).

The interviews were transcribed verbatim, and were analyzed based on the IPA interview data analysis procedures, which include (1) looking for themes in the first case (i.e., interview), (2) connecting the themes, (3) continuing the analysis with other cases (i.e., interviews), and (4) preparing the write-up [36] These procedures were not far off from Braun and Clarke's (2012) thematic analysis of qualitative data, which involves several phases, starting with familiarising oneself with the data, generalizing the initial codes, looking for the themes, revising potential themes, defining and naming themes, and producing a write-up or report [37]. Data validation in this study was conducted using Creswell's (2014) validity strategy, namely member checking to reduce bias and ensure the accuracy of the findings [38]. This was carried out by showing summaries of the findings to the interview participants and checking with them whether the findings accurately reflected their personal experiences [38].

## 3. Findings

### 3.1. Types of Social Media Used

The analysis revealed that the participants used several social media sites and applications (apps) on which they identified themselves as gay men. These sites and apps include the mainstream types (e.g., Twitter, Instagram, Facebook) and those that cater to gay men for social networking and dating purposes (e.g., Grindr, Bumble, Blued, Scruff). At least one participant, Arif, revealed that he used Clubhouse, the latest invitation- and audio-only social networking application. According to Clubhouse (2021), it is "a new type of social network based on voice—where people around the world come together to talk, listen and learn from each other in real-time" [39].

### 3.2. Thematic Analysis

The interview analysis produced four themes reflecting the participants' coming out strategies on social media: (1) I'm out and proud on social media; (2) I'm out, but I don't flaunt my pride; (3) I'm out and proud, and I'm out and discreet too; and (4) I'm out and discreet.

#### 3.2.1. I'm Out and Proud on Social Media

Only one participant, Azrul, reported that he was out and proud as a gay man on social media, mainly because of his active involvement in local LGBTQ activism. Azrul often used Twitter due to its user-friendly features that allowed him not only to tweet his current status and life updates but also to follow tweets from other users and connect with people who were important to him. This also explains Azrul's frequent use of Instagram, which afforded him a quick, easy, and convenient way to post current updates via infographics and Instagram stories. The following interview excerpt reveals Azrul's responses to the questions on whether he was out on social media and why he used social media as a gay man:

> Azrul: *I am very out and proud when it comes to my sexuality on all social media platforms whether it is Twitter, Facebook, Instagram, and so on, so I'm very out there. when your sexuality is like out there already when you're like when you're already out there I believe the comfort is there when your family knows . . . when your family knows that you're gay and that they're okay with it and so the whole world can know now . . . because your family is your last concern when it comes to these things I believe that was the what gateway to become open, whenever I tweet whatever I post they already know that's what I'm advocating for that's what I am . . .*

Azrul also reported that he was "very open" about his identity as a gay man on gay dating apps. This, however, very much depended on the nature, functions, and audience of these apps: Grindr for "hook(ing) up" and "sexual interaction"; Tinder for a more

meaningful conversation with other gay men; and Scruff for sexual interaction with specific preferred groups or "tribes" in the gay male community.

### 3.2.2. I'm Out, but I Don't Flaunt My Pride

Only one participant, Wang, stated that he was out as a gay man on social media and did not feel the need to be "loud about it" or to "show off [his] pride" in doing so. Wang cited several reasons, including the fact that many already knew about his identity and that he had the duty to maintain the status quo to preserve relations with family members, friends, and business associates. Wang explained that he used Facebook as the main social media platform to connect with his friends, followers, and business partners because of his status in the local entertainment industry. Wang explained that there was no pressure for him to come out or identify himself as gay on Facebook or post status updates and images that were related to his identity. Wang also reported his use of gay dating apps such as Grindr, Blued, and Jack'd "just for fun", with no serious intent. In other words, Wang was curious to know men who were available for hook-ups and would chat and fool around with them without having to meet them offline for sexual activities. He described being out on these apps using his real name and self-photo and received many hookup messages that he found funny and bizarre. Furthermore, Wang described "just normal things" he did on social media, in which he would regularly update his Facebook by sharing selfies and group selfies, and photos taken from his shows. Such "normal things" not only worked to preserve family/friend/business relations but also emphasized normative participation across mainstream social media platforms.

The following interview excerpt elucidates how Wang described his self-disclosure on social media:

> Wang: *I'm out but I don't show off my pride on social media or even in outside social media. I don't see any need for me to be loud about it because, because many people already know who I am even my business partners and they already accept it . . . that's why what I put on my social media doesn't have to be about my gay identity or things that are gay, just normal things that I share with people like family, friends, business partners and these I think is more important to maintain good relations with them.*

### 3.2.3. I'm Out and Proud, and I'm Out and Discreet Too

Two participants, Dennis and Jad, reported that they were not only out and proud as a gay man on social media, but also out and discreet about it. These dual strategies of coming out depended upon the nature, functions, and audiences of the social media platforms they used. Dennis, for example, owned two separate Facebook accounts: one for the "straight" audience, and another one for the "non-straight" audience. He explained that being gay on social media was more about expressing his identity (rather than coming out) without being constrained to do so, especially on his non-straight Facebook account—a place that was filled with those who already knew who he was even before the account was created. This was agreed with by Jad, who also created two separate Facebook accounts for two different audiences. Having these separate accounts enabled Jad to connect with people who may or may not know his identity with comfort, ease, and enjoyment, because "no one will get upset in the process". Both Dennis and Jad admitted to being "very" out and open on gay apps, and they did so based on the nature, functions, and audiences of these apps: Grindr for "hookups" and Tinder for "actual" or "serious" same-sex relationships with other gay men. The following interview excerpts illustrate Dennis and Jad's social media use as gay men:

> Dennis: *I'm in the middle between Out and Proud, and Out and Discreet . . . I think it really depends on what social media I'm in, what social (er) issue or what topics that I'm talking about because if you know like Grindr or whatever everyone is gay so there's no filter and there's not much filtering on that one too, but in terms of the main social media it's really depending on the topics that I'm talking about . . . so if let's say I'm talking about the gay community or gay culture that we already know, and that may be I*

*will filter . . . and when talking something about okay. okay like Thailand legalizes gay marriage then I will put that on to everyone . . . I'm not afraid to say I support this cause without telling that I'm gay.*

Jad: *I can say that I'm both, er both the out and proud and, and also the out and discreet. It really depends on the social media platforms that you use . . . you know, you have one for straight people, and another one for those who already know who you are . . . it's easy that way because no one will get upset in the process, I mean your family or friends who follow you on the straight social media, they don't have to know because you don't disclose yourself there, and on the gay social media you can express yourself openly . . . so it's really about how you manage yourself and your, your identity as a gay man in the social media platforms you're using.*

### 3.2.4. I'm Out and Discreet

Two participants, Nas and Arif, reported that there were out and discreet on social media, and that this was so for several reasons. Nas, on the one hand, used social media to connect with people and to keep himself updated with the latest news, especially those related to experiences of being gay. Nas stated that he did not use social media such as Facebook and Instagram, despite admitting that he used TikTok almost constantly to indirectly express his gay identity using personalized short videos. Nas also admitted that he was more discreet than out. He also stated that he did not put a real profile picture on the gay dating apps he regularly used because he was so sure of the possibility of not meeting any gay men there. The following interview excerpt illustrates Nas's views on his use of social media for self-disclosure:

Nas: *I use Facebook, Instagram, and TikTok. I mainly use these social media platforms just to be informed of any news related to being a gay man in Malaysia or just to stay connected with my friends. I don't really, openly, or actively share this because I tend to keep things to myself, and I like to stay discreet as much as I can.*

On the other hand, Arif reported that he was an out gay man to his circle of friends, but would remain discreet on social media such as Instagram due to his fear of upsetting and being disowned by his family members and relatives. He admitted that some gay men would be loud and proud about themselves on social media, but he did not see himself as such: he did not support the same cause despite being a gay man himself and being part of the gay community. The following interview excerpt reveals Arif's views on his use of social media for self-disclosure:

Arif: *Kenapa I guna sosial media as gay man I would say that I just want to socialize making friends macam tu. contoh macam, macam Instagram pun I cannot be openly gay sebab ada family ada kakak I like they are following me macam tu. if I openly gay like one day kat Facebook. so dekat Facebook kan ramai macam cousin makcik I semua yang friends dengan I so I would say lah it would affect my life lah I mean my family for them this is like a taboo and if I'm openly definitely diaorang takan mengaku I keluarga* [Translation: why I use social media as a gay man I would say that I just want to socialize, making friends like that . . . for example like Instagram I cannot be openly gay because my family my sisters like they are following me like that . . . if I openly gay like one day on Facebook . . . so on Facebook a lot of my cousins and aunts who are my friends so I would say it would affect my life I mean my family for them this is like a taboo and if I'm openly gay definitely they will disown me].

This was contrary to the way he put himself out as a gay man in Clubhouse—a place where most users were more open and accepting of his identity compared to those on Facebook. Arif also stated that he did not put up his real profile picture on gay dating apps such as Grindr because of his personal experience of not getting hit on by gay men there. To put it in another way, no gay men on these apps had shown sexual interest in Arif once he revealed his actual profile photo after a few seconds of chatting.

*3.3. Views on Social Media in Gay Men's Coming Out Processes*

The analysis revealed that the participants held different views regarding the important role of social media in the coming out process for gay men in Malaysia. For Nas, coming out online (and even offline) may not be such a good idea because being gay in Malaysia is "scary". This was agreed with by Arif, who stated that the fear was real given that being gay remained a taboo subject in this country, and that being gay was wrong in Islam and other religions. He knew homosexuality was wrong and did not support it although he was gay himself. For Arif, one should accept the fact that being gay was wrong and do so through "*muhasabah diri*" (self-judgment). However, for Dennis, Wang and Jad, coming out on social media was a matter of one's personal choice, and very much depended on one's feeling of ease, comfort, and safety in doing so. If one was not afraid of the consequences of going against the norms, then one should come out and express one's identity as a gay man on social media. This was agreed with by Azrul, who, in his position as an LGBTQ activist, provided some advice on coming out on social media and coming out in general. Azrul stated that one should not come out due to or under pressure, because one's sexuality belongs to oneself and not to others. He went on to say that people tended to be discreet for their own safety, career, and other reasons. If one decided to do so, one should tell one's closest friends, allies, and communities who could ensure and guarantee one's safety. The following interview excerpt shows Azrul's response to social media and coming out for gay men in Malaysia:

> Azrul: *I understand fully and I completely respect people to become discreet or become, to stay in the closet because in Malaysia you know it is your safety and your sexuality belongs to you it doesn't belong to anyone else . . . one doesn't have to know your sexuality or invalidate it . . . so whether you're out or discreet you know I respect both and so for my aspect is that because I am an out gay man in Malaysia, I believe that I have a voice that can channel injustices when it come s to LGBTQ issues through social media such as Twitter . . .*

## 4. Discussion

Coming out is a big part of many young gay men's lives. In today's digital world, an increasing number of gay male youths have taken to social media sites to disclose or embrace their sexuality in a variety of ways using various strategies. The present study examined coming out strategies on social media among young gay men in Malaysia, where homosexuality remains invalidated and illegalized. More specifically, the study investigated the types of social media used by these gay male youths, the strategies they used to disclose their identity on social media, and their views on coming out on social media.

Our results indicated that gay male youths in the country (particularly those interviewed in the study) used various social media platforms for self-disclosure; these platforms can be divided into two types: mainstream social media and gay social media. The results are consistent with previous studies—including those by Hanckel, Vivienne, Byron, Robards, and Churchill (2019) and Syahputra and Yuliana (2016)—that revealed the myriad of social media platforms that their LGBTQ participants used for outness to family and others, including Facebook, Instagram, Twitter, and Jack'd. Our study builds on existing research on social media experience among LGBTQ individuals in Malaysia in that Twitter is one but not the only avenue through which LGBTQ individuals assert and express their sexual identities.

The current study found four strategies that young gay Malaysians used for self-disclosure on social media: out, and proud; both out and proud, and out and discreet; out and discreet; and social media closeted. This contrasts with three strategies found in Owen's (2017) study, mainly because of the current sample. That is, young gay men in Malaysia (particularly the participants in our study) can take up a dual strategy of self-disclosure on social media, consisting of being proud and out on social media sites where users are more open and accepting of their gay identity, and being out and discreet on social media platforms to prevent the repercussions of being found out publicly by family

members, friends, and relatives. Finally, our results are also consistent with the two broad streams of literature in the field of coming out on social media: first, social media is a safe place for self-disclosure among LGBT youth [26,28], and second, social media is "effectively unsafe" for the same purpose [40]. While a couple of our participants stated that coming out as gay on social media was possible if one felt safe and comfortable, without having the pressure to do so, the rest thought otherwise. This is because they were concerned with the fact that homosexuality is not accepted within society in Malaysia.

However, despite the similarities and differences found between our study and those we cited here, it is important to make a brief note on (1) the broader considerations of the impact of 'outness', (2) the sociological realm of social media use, and (3) activism through online self-disclosure among gay men in Malaysia

First, the findings reveal that family and friends, as well as ties (or the lack thereof) to the community, can have an impact on the participants' capacity to self-disclose. A majority of the participants were able to disclose their identity with their close circle of friends, unlike their 'straight' peers, family members, and the larger community. This is due to several reasons, such as the participants' concerns of being found out, the fear of being discriminated against, and having their identity stigmatized because homosexuality is widely rejected in Malaysian society.

Second, the findings provide some broad insights into the sociological realm of the use of social media among gay men in Malaysia. As discussed earlier, our gay male participants were able to disclose their sexual orientation (albeit to varying degrees) not only on mainstream social media platforms but also on those that are designed specifically for gay men. This is mainly because the participants were able to access gay-oriented networking websites (e.g., Grindr) that have been banned in other countries (e.g., Indonesia, Lebanon, Qatar, Saudi Arabia, Turkey, and the UAE) on the basis that they posed danger to national security [41]. However, this does not mean that the government of Malaysia has been mum on the matter. It has been reported that, since 2018, the Malaysian Communications and Multimedia Commission (MCMC) has blocked a total of 2484 pornographic sites in its efforts to curb websites that promote pornography and lesbian, gay, bisexual, and transsexual (LGBT) culture [42]. A survey conducted by the Pelangi Campaign in 2018 revealed that 54.62% of LGBTIQ respondents experienced difficulty accessing LGBT websites such as www.planetromeo.com (accessed on 31 January 2022), www.gaystarnews.com (accessed on 31 January 2022), www.utopia-asia.com (accessed on 31 January 2022), and www.ilga.com (accessed on 31 January 2022) [43]. Nonetheless, LGBTQ individuals in Malaysia continue to seek new ways in which they can construct and express their identities on social media, given the great opportunities afforded by the Internet in terms of connectivity and the ease and freedom to do almost anything that the real, physical world cannot offer.

Third, the findings support Erving Goffman's theory of self-representation, which has foregrounded much research on online identity management on social media. The theory is useful in our study, as it has enabled us to see online identity management among gay male social network users. Specifically, our participants created their online identities or personal façade by disclosing and managing these identities through different strategies for different purposes. These include (1) concealing their gay identity by creating a specific social media account for the straight audience and being cautious when posting about the homosexual aspect of their personal life, and (2) disclosing their gay identity on gay dating apps by putting up profile pictures and stating specific dating preferences to hook up with other gay men. Such online management strategies are consistent with those discussed by Duguay (2016) in terms of how LGBTQ individuals negotiate sexual identity disclosure on mainstream social media platforms, and Merunková and Šlerka (2019) in terms of the different goals of online identity management among social media users in general: (1) the exploration and development of their identities, which are facilitated by social media; (2) the concealment of authentic selves on social media for impression management; (3) the protection of their identities on social media to manage multiple selves; and (4) the expression of authentic selves on social media.

Finally, our findings reveal insights into queer activism, particularly into how the participants participate in advocacy in both open and discreet ways and manage these boundaries across social media spaces. We can see this in Azrul, who is not only out and proud as a gay man both online and offline but also uses his social media platform as an activist to promote awareness about the LGBTQ community in Malaysia. Dennis, on the other hand, is also involved in LGBTQ advocacy in discreet ways by, for example, showing support for gay rights activism on social media without necessarily coming out or identifying himself as a gay man. We believe that these different advocacy participations extend Mohamad Tuah and Mazlan's (2020) finding that LGBTQ-friendly cliques or virtual support groups are one but not the only means through which LGBTQ individuals in Malaysia can take up the cause of LGBTQ advocacy.

## 5. Conclusions

This study contributes to the research on coming out strategies among gay youths, intending to address the knowledge gap regarding gay male youths in Malaysia and their self-disclosure techniques on social networking sites and apps. This study also provides insights into the role that social media play in coming out for gay male youths in Malaysia, in which homosexuality is not accepted by a large percentage of the Malaysian public [8,9]. While social media provide young gay Malaysian men with a venue in which they can disclose their sexual identity in different ways for different purposes and audiences, social media may not necessarily be a safe space for self-disclosure, given their attendant adverse consequences. Our findings suggest that more needs to be known regarding the extent to which social media can assist gay male youths in the development of their wellbeing, especially to accomplish the developmental tasks of constructing identity and coming to terms with their sexuality. This study is not without limitations. A major limitation was the small number of gay men who agreed to participate in the study. This presented an obstacle to the production of a generalization from the sample. The results of this study would have been greater or richer with larger sample size. It is appropriate to say that this present study could serve as a pilot for further research. Because the study focused on gay male youths from two cities in Malaysia, considerations should be made by future researchers to include young gay men from across the country. This inclusion would provide a better understanding of social media and the role they play in their coming out processes.

**Author Contributions:** Conceptualization, C.J. and A.J.b.A.H.; methodology, C.J.; validation, C.J. and A.J.b.A.H.; investigation, C.J. and A.J.b.A.H.; writing—original draft preparation, C.J. and A.J.b.A.H.; writing—review and editing, C.J. and A.J.b.A.H.; funding acquisition, C.J. All authors have read and agreed to the published version of the manuscript.

**Funding:** This research was funded by the Ministry of Education of Malaysia under the Fundamental Research Grant Scheme, grant number F09/FRGS/1874/2019.

**Institutional Review Board Statement:** The study was conducted according to the guidelines of the Declaration of Helsinki and approved by the Institutional Review Board (or Ethics Committee) of Universiti Malaysia Sarawak (Protocol Code: HREC(BP)2020(1)/02 (approval date: 8 December 2020)).

**Informed Consent Statement:** Informed consent was obtained from all of the subjects involved in the study.

**Data Availability Statement:** The data presented in this study are available on request from the corresponding author. The data are not publicly available due to ethical concerns.

**Acknowledgments:** We would like to thank our participants, without whom this study would not have been accomplished.

**Conflicts of Interest:** The authors declare no conflict of interest.

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
