# Peer review of "Coming Out Strategies on Social Media among Young Gay Men in Malaysia"

_2673-995X, doi:10.3390/youth2010004_

Round 1
Reviewer 1 Report
This study examines the coming out strategies of gay young men on social media, drawing on 6 in-depth interviews with young gay men in Malaysia. Overall, this an interesting study – it is however missing some empirical studies, which would benefit this paper (which I have included below), and the author(s) need to better integrate the theoretical framework into the paper overall. This will strengthen the manuscript. I propose, subject to the major revisions suggested it is accepted.
I have added comments below -
Abstract – The current framing is that there is much from western countries, and little from non-western. Given the scope of the study it would be more appropriate to frame the gap as limited data on the social media exp's of gay men in Malaysia.
Introduction – first para - overall needs a link (last sentence?) to the aim/focus of this study.
P.1, Line 31-32 – needs a reference.
p.1 Lines 34 – 37 – a bit vague, please include examples of factors that foster coming out and the consequences; it would also be useful to include the varied difficulties different people with different intersections have coming out (particularly in the context of this paper – i.e. Malaysia).
p.1 Line 39 – include a definition of social media – confirm that it includes hook-up/dating apps. Additionally include a definition of affordances, as the authors use this term to make sense of and explain the experiences of their participants.
P.2 – the primary focus in examples is on Facebook – I propose the author(s) consider examples outside of Facebook use as well to illustrate the diverse uses of social media platforms they are discussing.
p.2 Lines 87-88 – the strategies need explaining to the reader – I propose the author(s) paraphrase these instead.
p.3 Lines 102 – 103 – this needs linking back to this paper. Explain the gaps in the literature that are informing this study.
P.3-4 – theory of identity construction. This is interesting but you need to explain how it fits into this study – are you using the idea of the ‘façade’ to make sense of the experiences of the young people in this study - if so tell this to the reader, and use it in later stages of the paper. The theory is interesting and reminds me of research on polymedia use – (Madianou, and Miller, 2013) as well as work on ‘curation’ (Hanckel et al, 2019 (cited)), which could be useful to include here. The model is particularly useful in this study as it points to the varied contexts of use, which this study surfaces in relation to Malaysia.
Methods –
- P. 4 – explain how recruitment was undertaken
- P. 5 - include IRB project number and university;
- P. 5 – lines 207 – 208 – indicate themes, and explain these will be discussed in the findings section.
Findings –
- P. 6 Lines 243 – 244 – what does ‘no serious intent’ mean. This needs explanation as before you indicated that sometimes these apps are used just for knowledge acquisition etc.
- P. 6 lines 251 -252 – ‘just normal things’ – what are these? Can the author(s) explain? Did the interviewee provide more detail and explain how normal (read as 'not gay' here) worked to preserve family/friend/business relations? Does this, for instance, emphasise a normative participation across platforms?
- P. 7 – Lines 290 – 292 – explain TikTok video use here in more detail.
- P. 8 – Lines 321 – 323 need some further explanation – I don’t understand ‘not getting hit on…’
- Was there data on digital literacy and/or limitations to internet access/accessibility – did this play a role in how they used these spaces?
Discussion -
- Missing in the discussion section are broader considerations of impact of ‘outness’ on family/friends. Also how do the ties (or lack thereof) to community also impact on the ‘outness’ of participants online. This emerges in your findings, and would benefit from inclusion in the discussion.
- The discussion brings in additional studies, which need to be referenced earlier in the beginning of the manuscript (p. 8 -9 , Lines 360 – 364).
- P. 9, Lines 367 – 369 – this is vague – what is this study offering, and how is it building on existing research.
- There is no engagement with the theoretical ideas presented earlier around the ‘façade’ – how is this theory/framework useful here, what has it helped the author(s) see/surface through this study? Does this study build on this work. This needs to be discussed here.
- Also there are clear moments of activism or queer world-making/life making in the findings – the authors could include a discussion about how these gay men participate in advocacy in open and discreet ways, and manage these boundaries across social media spaces.
Overall the manuscript needs some editing – it is missing close brackets, missing full stops, names of authors names referenced in text at times misspelt (i.e. Hankel should be Hanckel), and the tense used is not always correct.
Additional references to engage with -
- Cho, A (2017) Default publicness: queer youth of color, social media, and being outed by the machine. New Media & Society 20(9): 3183–3200.
- Duguay, S (2014) ‘He has a way gayer Facebook than I do’: investigating sexual identity disclosure and context collapse on a social networking site. New Media & Society 18: 891–907.
- Hanckel B, Chandra S (2021) Social media insights from sexuality and gender diverse young people during COVID-19. Western Sydney University, Sydney. https://doi.org/10.26183/kvg0-7s37
- Madianou, M. and Miller, D. (2013) ‘Polymedia: Towards a new theory of digital media in interpersonal communication’, International Journal of Cultural Studies, 16(2), pp. 169–187
- Wei (2021): Out on YouTube: queer youths and coming out videos in Asia and America, Feminist Media Studies
- Zhang W (2013) Redefining youth activism through digital technology in Singapore. International Communication Gazette 75:253–270.
Author Response
Reviewer 1
This study examines the coming out strategies of gay young men on social media, drawing on 6 in-depth interviews with young gay men in Malaysia. Overall, this an interesting study – it is however missing some empirical studies, which would benefit this paper (which I have included below), and the author(s) need to better integrate the theoretical framework into the paper overall. This will strengthen the manuscript. I propose, subject to the major revisions suggested it is accepted.
I have added comments below -
Abstract – The current framing is that there is much from western countries, and little from non-western. Given the scope of the study it would be more appropriate to frame the gap as limited data on the social media exp's of gay men in Malaysia.
Author response: We agree with this suggestion. Accordingly, we have reframed the gap and rewritten the abstract to highlight the limited data on the social media experiences among gay men in Malaysia. Kindly see lines 4-8.
Introduction – first para - overall needs a link (last sentence?) to the aim/focus of this study.
Author response: Thank you for pointing this out. We have added the link as per the reviewer’s suggestion. Kindly see the following sentence (page 1):
“The present study aims to add this discussion by examining coming out strategies on social media among young gay men in Malaysia.”
P.1, Line 31-32 – needs a reference.
Author response: Thank you for pointing this out. We have added a reference as per the reviewer’s suggestion. Kindly see [1] (page 1).
p.1 Lines 34 – 37 – a bit vague, please include examples of factors that foster coming out and the consequences; it would also be useful to include the varied difficulties different people with different intersections have coming out (particularly in the context of this paper – i.e. Malaysia).
Author response: We thank the reviewer and agree with the assessment. We have added the examples and additional information as per the reviewer's suggestion. Kindly see lines 41-50.
p.1 Line 39 – include a definition of social media – confirm that it includes hook-up/dating apps. Additionally include a definition of affordances, as the authors use this term to make sense of and explain the experiences of their participants.
Author response: Thank you for pointing this out. We have added a reference as per the reviewer’s suggestion. Kindly see lines 52-67.
P.2 – the primary focus in examples is on Facebook – I propose the author(s) consider examples outside of Facebook use as well to illustrate the diverse uses of social media platforms they are discussing.
Author response: Thank you for pointing this out. We believe that the primary focus on Facebook ties in well with our study, given that our participants use it as their main social media. However, we are willing and receptive to accept further corrections.
p.2 Lines 87-88 – the strategies need explaining to the reader – I propose the author(s) paraphrase these instead.
Author response: Thank you for pointing this out. We have made the necessary changes. Kindly see lines 131-140.
p.3 Lines 102 – 103 – this needs linking back to this paper. Explain the gaps in the literature that are informing this study.
Author response: Thank you for pointing this out. We have made the necessary changes. Kindly see lines 190-193.
P.3-4 – theory of identity construction. This is interesting but you need to explain how it fits into this study – are you using the idea of the ‘façade’ to make sense of the experiences of the young people in this study - if so tell this to the reader, and use it in later stages of the paper. The theory is interesting and reminds me of research on polymedia use – (Madianou, and Miller, 2013) as well as work on ‘curation’ (Hanckel et al, 2019 (cited)), which could be useful to include here. The model is particularly useful in this study as it points to the varied contexts of use, which this study surfaces in relation to Malaysia.
Author response: Thank you for pointing this out. We created this framework based on our understanding of Goffman’s theory and also the four themes that emerged from the study by Talbot, Talbot, Roe, and Briggs (2020) [30]. We found the research on polymedia and curation useful and would include them in our future studies on social media experiences among LGBTQ individuals in Malaysia
Methods –
- 4 – explain how recruitment was undertaken:
Author response: Thank you for pointing this out. An explanation of the recruitment has been added as per the reviewer’s request. Kindly lines 226-243.
- 5 - include IRB project number and university;
Author response: Thank you for pointing this out. The details have been added as per the reviewer’s request. Kindly see lines 248-256.
- 5 – lines 207 – 208 – indicate themes, and explain these will be discussed in the findings section.
Author response: Thank you for pointing this out. The themes have been added as per the reviewer’s request. Kindly see lines 306-309.
Findings –
- 6 Lines 243 – 244 – what does ‘no serious intent’ mean. This needs explanation as before you indicated that sometimes these apps are used just for knowledge acquisition etc.
Author response: Thank you for pointing this out. An explanation on ‘no serious intent’ has been added as per the reviewer’s request. Kindly see lines 342-344.
- 6 lines 251 -252 – ‘just normal things’ – what are these? Can the author(s) explain? Did the interviewee provide more detail and explain how normal (read as 'not gay' here) worked to preserve family/friend/business relations? Does this, for instance, emphasise a normative participation across platforms?
Author response: Thank you for pointing this out. An explanation on ‘just normal things’ has been added as per the reviewer’s request. Kindly see lines 346-350.
- 7 – Lines 290 – 292 – explain TikTok video use here in more detail.
Author response: Thank you for pointing this out. Unfortunately, the participants did not really provide further descriptions of the TikTok video use.
- 8 – Lines 321 – 323 need some further explanation – I don’t understand ‘not getting hit on…’
Author response: Thank you for pointing this out. An explanation on not getting hit on has been added as per the reviewer’s request. Kindly lines 429-430.
Was there data on digital literacy and/or limitations to internet access/accessibility – did this play a role in how they used these spaces?
Author response: Thank you for pointing this out. Unfortunately, there was no data on this, but we were aware of the fact that all participants were able to use social media with ease (e.g., basic features of updating posts and photos, communicating/interacting with others) and had no limitations to internet access.
Discussion -
Missing in the discussion section are broader considerations of impact of ‘outness’ on family/friends. Also how do the ties (or lack thereof) to community also impact on the ‘outness’ of participants online. This emerges in your findings, and would benefit from inclusion in the discussion.
Author response: Thank you for pointing this out. An explanation of the broader considerations of impact of ‘outness’ has been added per the reviewer’s request. Kindly lines 493-499.
The discussion brings in additional studies, which need to be referenced earlier in the beginning of the manuscript (p. 8 -9 , Lines 360 – 364).
Author response: Thank you for pointing this out. We have removed these additional studies and referenced only the ones we mentioned at the beginning of the manuscript for consistency.
- 9, Lines 367 – 369 – this is vague – what is this study offering, and how is it building on existing research.
Author response: Thank you for pointing this out. An explanation of what the study is offering and how it builds on existing research has been added as per the reviewer’s request. Kindly see lines 468-473.
There is no engagement with the theoretical ideas presented earlier around the ‘façade’ – how is this theory/framework useful here, what has it helped the author(s) see/surface through this study? Does this study build on this work. This needs to be discussed here.
Author response: Thank you for pointing this out. An explanation of the theoretical ideas has been added as per the reviewer’s request. Kindly see lines 518-533.
Also there are clear moments of activism or queer world-making/life making in the findings – the authors could include a discussion about how these gay men participate in advocacy in open and discreet ways, and manage these boundaries across social media spaces.
Author response: Thank you for pointing this out. An explanation of moments of activism has been added as per the reviewer’s request. Kindly see lines 534-544.
Overall the manuscript needs some editing – it is missing close brackets, missing full stops, names of authors names referenced in text at times misspelt (i.e. Hankel should be Hanckel), and the tense used is not always correct.
Author response: Thank you for pointing this out. We have corrected the misspelt name and other grammatical and writing errors as per the reviewer’s request. However, we are willing and receptive to accept further corrections.
Additional references to engage with -
- Cho, A (2017) Default publicness: queer youth of color, social media, and being outed by the machine. New Media & Society 20(9): 3183–3200.
- Duguay, S (2014) ‘He has a way gayer Facebook than I do’: investigating sexual identity disclosure and context collapse on a social networking site. New Media & Society 18: 891–907.
- Hanckel B, Chandra S (2021) Social media insights from sexuality and gender diverse young people during COVID-19. Western Sydney University, Sydney. https://doi.org/10.26183/kvg0-7s37
- Madianou, M. and Miller, D. (2013) ‘Polymedia: Towards a new theory of digital media in interpersonal communication’, International Journal of Cultural Studies, 16(2), pp. 169–187
- Wei (2021): Out on YouTube: queer youths and coming out videos in Asia and America, Feminist Media Studies
- Zhang W (2013) Redefining youth activism through digital technology in Singapore. International
Author response: We thank the reviewer for the reading list and have found the work by Duguay (2016) as well-suited for our study.
Reviewer 2 Report
Comments and suggestions:
1 Explain where the two major cities in Malaysia are. Is it in Kuala Lumpur or somewhere? It is very important to know at least the demography of 'gay cities' in Malaysia.
2. Are the names of the people using pseudonyms? Is using pseudonyms, a state in the paper that the names given are pseudonyms.
3. Can the 'discussion' part of the problem be widened a little more? I see it is good, but what I see is the problem presented is only a problem that is almost the same as the title of the paper. I suggest looking at the sociological realm of the use of social media among gay men in Malaysia, such as whether there are social media banned in Malaysia like in Indonesia.
Author Response
Reviewer 2
Comments and suggestions:
1 Explain where the two major cities in Malaysia are. Is it in Kuala Lumpur or somewhere? It is very important to know at least the demography of 'gay cities' in Malaysia.
Author response: We thank the reviewer and agree with the assessment. Accordingly, we have added the explanation on the two major cities in Malaysia and provided some details on the demography of the country’s ‘gay cities’. Kindly see lines 226-236.
- Are the names of the people using pseudonyms? Is using pseudonyms, a state in the paper that the names given are pseudonyms.
Author response: Thank you for pointing this out. We used pseudonyms instead of the participants’ real, actual names to ensure their confidentiality and anonymity, which is in accordance with many previous and current studies of LGBTQ individuals. Therefore, we have included an explanation on the pseudonyms. Kindly see lines 240-242.
- Can the 'discussion' part of the problem be widened a little more? I see it is good, but what I see is the problem presented is only a problem that is almost the same as the title of the paper. I suggest looking at the sociological realm of the use of social media among gay men in Malaysia, such as whether there are social media banned in Malaysia like in Indonesia.
Author response: We agree with the reviewer’s assessment. We have widened the discussion by elaborating the sociological realm of the use of social media among gay men in Malaysia (e.g., whether there are social media banned in Malaysia). Kindly see lines 500-517.
Reviewer 3 Report
111 – not Islam, it is not monolithic. Maybe choose a more nuanced expression such as “mainstream conservative Islam” or “Malaysian Sharia” or even “Malaysian representation of Sharia laws” (which could also lead to a more precise description of Malaysian mainstream Islam with the very interesting references in 15, 16 and 17). Plus you are very well describing later (in 112 to 117) how that “Sharia” evolved to become more conservative lately.
384 – very important determining factor. I would love to hear more in a future study about the reasons why it is happening now.
Nothing to add: very well structured article, interesting first results.
Author Response
Reviewer 3
111 – not Islam, it is not monolithic. Maybe choose a more nuanced expression such as “mainstream conservative Islam” or “Malaysian Sharia” or even “Malaysian representation of Sharia laws” (which could also lead to a more precise description of Malaysian mainstream Islam with the very interesting references in 15, 16 and 17). Plus you are very well describing later (in 112 to 117) how that “Sharia” evolved to become more conservative lately.
Author response: Thank you for pointing this out. As suggested by the reviewer, we have replaced the word ‘Islam’ with “Malaysian representation of Sharia laws”, which ties in well with the description of Malaysian mainstream Islam that is referenced by 15, 16 and 17. Kindly see lines 162-168.
384 – very important determining factor. I would love to hear more in a future study about the reasons why it is happening now.
Author response: We think this is an excellent suggestion. We believe it is worth taking into consideration in future research. Once again, thank you!
Nothing to add: very well structured article, interesting first results.
Author response: Thank you!